

**Estimation of pollen counts from light scattering intensity when sampling multiple pollen taxa** —
**Establishment of Automated Multi-taxa Pollen Counting Estimation System (AME System)**—
Kenji Miki[1*], Shigeto Kawashima[1]
[1] Graduate School of Agriculture, Kyoto University, Oiwake-cho, Kitashirakawa, Sakyo-ku, Kyoto 606-8502,
Japan
*Correspondence to*: Kenji Miki (kmiki@elsi.jp)
**Abstract.**
Laser optics have long been used in pollen counting systems. To clarify the limitations and potential new
applications of laser optics for automatic pollen counting and discrimination, we determined the light scattering
patterns of various pollen types, tracked temporal changes in these distributions, and introduced a new theory for
automatic pollen discrimination. Our experimental results indicate that different pollen types often have different
light scattering characteristics, as previous research has suggested. Our results also show that light scattering
distributions did not undergo significant temporal changes. Further, we show that the concentration of two
different types of pollen could be estimated separately from the total number of pollen grains by fitting the light
scattering data to a probability density curve. These findings should help realize a fast and simple automatic pollen
monitoring system.






## 1 Introduction

Pollen counting is a time-consuming and labor-intensive task that requires professional skills. However, recent technological developments have made automatic pollen sampling and identification possible (Buters et al. 2018), for example, with recognition systems using microscopic images of pollen grains (Boucher et al. 2002; Ranzato et al. 2007; Oteros et al. 2015), pollen color patterns from pollen images (Landsmeer et al. 2009), fluorescence emission signals, (Swanson and Huffman 2018; Mitsumoto et al. 2009; Mitsumoto et al. 2010; Richardson et al. 2019), light scattering (Crouzy et al. 2016; Šaulienė et al. 2019, holographic images (Sauvageat et al. 2019), size and morphological characteristics (O'Connor et al. 2013), real-time PCR (Longhi et al. 2009), texture and infrared patterns of microscopic images of pollen (Marcos et al. 2015; Gottardini et al. 2007; Chen et al. 2006), or a combination of several of these. Many studies applied machine learning algorithms to the problem (Punyasena et al. 2012; Tcheng et al. 2016; Crouzy et al. 2016; Gonçalves et al. 2016; Gallardo-Caballero et al. 2019; Šaulienė et al. 2019). These automated pollen identification methods have been applied not only to aerobiological research but also to palynological studies for the identification of fossilized pollen (France et al. 2000; Kaya et al. 2014; Li et al. 2004; Zhang et al. 2004; Rodríguez-Damián et al. 2006).

Analysis using light scattering patterns has a particular focus, with several methods being developed for establishing an automatic aerosol or bioaerosol counting system (Huffman et al. 2016). For example, polarization signals can be used to discriminate *Cryptomeria japonica* from polystyrene spherical particles (Iwai 2013). Studies applying machine learning algorithms have shown that light scattering patterns can be used for automatic classification and counting of multiple pollen taxa simultaneously (Crouzy et al., 2016;  Sauliene et al., 2019). Other studies have  applied statistical techniques to compare the light scattering data and number of multiple taxa pollen grains (Kawashima et al. 2007, 2017; Matsuda and Kawashima 2018). Surbek et al. (2011) also studied the discrimination method for Hazel, Birch, Willow, Ragweed, and Pine pollen showing that they have distinct characteristics in the backward and sideward light scattering patterns.

In the present study, light scattering patterns from various pollen taxa are investigated with a KH-3000 to verify whether they have different light scattering patterns. A novel method is also proposed to discriminate between two taxa with similar scattering patterns.

## 2 Materials and methods

A protection cylinder (radius = 5 cm, height = 30 cm) was attached to the sampling tube of a KH-3000-01 laser-optics-based automatic pollen counter (Yamatronics, Japan). The KH-3000-01 is a widely used automatic pollen counting system (e.g. Wang et al. 2014; Takahashi et al. 2001; Miki et al. 2017, 2019; Kawashima et al. 2007, 2017; Matsuda and Kawashima 2018). A laser irradiates particles that pass through the sampling system and the forward and side scattering signals from each particle are recorded. In this study pollen grains from known taxa were injected through an injection tube in the wall of the protection cylinder and sampled in the KH-3000-01. The side and front scattering intensities were evaluated by converting the light intensity into a voltage.

### 2.1 Temporal changes in light scattering patterns

*Alnus* pollen grains were directly sampled from catkins on a tree growing at the Swiss Federal Office of Meteorology and Climatology on a sunny morning on February 28 2019. Light scattering measurements were taken using the fresh pollen grains soon after they were collected. The remaining pollen grains were stored in tubes and scattering patterns were reevaluated after storing them for 1 h, 2 h, 6 h, and 10 days. Multiple comparisons using the Bonferroni method were performed on the side and front scattering data to assess whether the light scattering distributions showed changes after storage. Bonferroni method is a multiple comparison method used for non-parametric data sets. In order to carry out the multiple comparison, 316 scattering data of each taxa were picked up because the Bonferroni method requires the same amount of data of each taxa and 316 scatteing data was the smallest amount of data amongst each time step (10 day).

### 2.2 Light scattering patterns of different pollen taxa

Dried pollen grains from *Alnus*, *Ambrosia*, *Artemisia*, *Betula*, *Castanea*, *Cedrus*, *Corylus*, *Fagus*, *Fraxinus*, *Helianthus*, *Olea*, *Phleum*, *Quercus*, *Taxus*, and *Zea* were sampled in a similar way. These taxa are representative of the pollen types commonly observed in Europe. After collecting the light scattering distributions of each pollen type, multiple comparisons using the Bonferroni method were performed to evaluate whether these distributions differ significantly from each other. In order to carry out the multiple comparison, 210 scattering data of each taxa were picked up based on the smallest amount of data amongst the taxon (*Helianthus*).

### 2.3 Automatic discrimination theory



Atmospheric
Measurement
Techniques



Discussions

To carry out simple and fast automatic pollen discrimination, the number of pollen grains of each type from the
total number of pollen grains was calculated as follows.
For two different types of pollen (A and B) in the side scattering intensity range $a - b$ and in the front scattering
intensity range $c - d$, the following equation holds:

$$\int_a^b P_{A_{side}(x)}\, dx = p_{A_{side}}$$
$$\int_a^b P_{B_{side}(x)}\, dx = p_{B_{side}}$$
$$\int_c^d P_{A_{front}(x)}\, dx = p_{A_{front}} \tag{1}$$
$$\int_c^d P_{B_{front}(x)}\, dx = p_{B_{front}}$$

where $P$ is the representative probability density function of the scattering intensity. $p$ is the representative
probability of the scattering intensity of each pollen grain lying in the integration intervals.
Next, the scattering intensity distribution that gives the number of pollen grains at each scattering intensity was
fitted to a distribution function. In this experiment, the normal distribution was fitted to the number of pollen
grains in every 100 mV steps. The gaussian function is written as:

$$f_{(x)} = \frac{\alpha}{\sqrt{2\pi}} \exp\left\{-\frac{(x-\mu)^2}{2\sigma^2}\right\} + c \tag{2}$$

where $\alpha$ and $c$ are coefficients, $\mu$ is the mean, $\sigma$ is the standard deviation.
Fitting the data to the normal distribution function enables one to calculate the probability of a pollen grain
showing a certain light scattering intensity. The probability density of the normal distribution function ($P$) is
written as:

$$P_{(x)} = \frac{1}{\sqrt{2\pi\sigma^2}} \exp\left\{-\frac{(x-\mu)^2}{2\sigma^2}\right\} \tag{3}$$

Fitting was performed by nonlinear optimization. The normal distribution was chosen so that we can handle the
light scattering plots using a known function.
Equation (1) gives

$$C_1 p_{A_{side}} N_A + C_2 p_{B_{side}} N_B = n_{side\ a-b}$$
$$C_3 p_{A_{front}} N_A + C_4 p_{B_{front}} N_B = n_{front\ c-d} \tag{5}$$
$$N_A + N_B = N_{total}$$

Here, $N$ is the number of sampled pollen grains of each pollen type, which are the values to be calculated. $N_{total}$
is the total number of sampled pollen grains and $n$ is the total number of sampled pollen grains in the integration
interval, which are known numbers. $C$ is the correction factor defined by the following equation:

$$C = \frac{\int_{-\infty}^{+\infty} P_{(x)} dx}{\int_0^{4500} P_{(x)} dx}$$
$$= \frac{1}{\int_0^{4500} P_{(x)} dx} \tag{6}$$

$C$ is needed for renormalization of the probability distribution because the device KH-3000-01 is able to detect
the scattering intensity only in the range of 0–4500mV.
By solving two equations in Eq. (5), $N_A$ and $N_B$ will be theoretically estimated.



In this paper, *Alnus* and *Artemisia* were chosen as examples to evaluate the usability of the theory above. Because
fitting worked well in the range of 600–800mV for the side scattering and 300–500mV for the front scattering,
$a = 600$, $b = 800$, $c = 300$ and $d = 500$ were substituted in Eq. (5) . The evaluation tests were carried out five
times using the light scattering data for both *Alnus* and *Artemisia* (Fig. 1).
The magnitude of the estimation error is calculated as follows.

$$error\ (\%) = \frac{|actual - estimation|}{actual} \times 100 \tag{7}$$



**3 Results**
**3.1 Temporal changes in light scattering pattern**
The scattering distribution of *Alnus* pollen (Fig. 2) showed no significant temporal changes in scattering
distributions in 10 day (Table 1).
**3.2 Light scattering distributions of different pollen taxa**
Pollen grains with smaller sizes tend to show smaller voltage values (Fig. 3).. The results of the multiple
comparisons (Table 2) indicated that there is always a significant different between side and front scattering
between two different pollen types except between:
Side scattering: *Alnus-Ambrosia, Alnus-Corylus, Alnus-Olea, Ambrosia-Fraxinus, Betula-Phleum, Betula-*
*Quercus, Corylus-Olea, Fagus-Zea, Artemisia-Fraxinus, Helianthus-Zea, Phleum-Quercus*
Front scattering: *Alnus-Corylus, Alnus-Quercus, Ambrosia-Artemisia, Ambrosia-Fraxinus, Artemisia-Fraxinus,*
*Betula-Phleum, Betula-Quercus, Castanea-Olea, Cedrus-Helianthus, Corylus-Quercus, Fagus-Helianthus,*
*Fagus-Zea, Phleum-Quercus*
**3.3 Automatic counting**
Counting the number of pollen grains of each type can be carried out by solving the two equations from Eq. (5),
side ($n_{side\ a-b}$) and front ($n_{front\ c-d}$), side ($n_{side\ a-b}$) and total ($N_{total}$) , front ($n_{front\ c-d}$) and total ($N_{total}$). The
parameters of the probability density curve of the side and the front (Fig. 4) light scattering distributions of *Alnus*
and *Artemisia* were estimated as follows:
$$P_{Alnus_{side}} : (\alpha, \mu, \sigma, c) = (433.58, 555.13, 223.85, 14.74)$$
$$P_{Alnus_{front}} : (\alpha, \mu, \sigma, c) = (588.98, 419.45, 192.67, 10.31)$$
$$P_{Alnus_{front}} : (\alpha, \mu, \sigma, c) = (600.25, 348.67, 159.96, 16.25)$$
$$P_{Artemisia_{front}} : (\alpha, \mu, \sigma, c) = (1028.57, 202.64, 107.32, 13.00)$$
The results (Fig. 5) show that the estimated number of pollen grains had average errors of 46.80%, 33.9%, 39,12%
for *Alnus* and 30.81%, 18.77%, 20.57% for *Artemisia* (Table 3).

**4 Discussion**
Temporal changes in the shapes of pollen grains are expected to affect the changes in light scattering patterns.
However, our experimental data indicate that light scattering patterns show little to no changes over time (up to
at least 10 days). Thus, there should be no problem using pollen grains that are either fresh or have been stored
for several days for studies with the KH-3000. Further investigation is required to understand whether this is true
for species other than *Alnus* and for longer periods of time. Understanding the morphological stability of each
pollen type would be helpful to understand the temporal stability of light scattering patterns.
Light scattering data from various pollen taxa indicate that it is not possible to discriminate between the side
scattering patterns of *Alnus* vs *Ambrosia*, *Alnus* vs *Corylus*, *Alnus* vs *Olea*, *Ambrosia* vs *Fraxinus*, *Betula* vs



*Phleum*, *Betula* vs *Quercus*, *Corylus* vs *Olea*, *Fagus* vs *Zea*, *Artemisia* vs *Fraxinus*, *Helianthus* vs *Zea*, *Phleum*
vs *Quecus* and the front scattering patterns between *Alnus* vs *Corylus*, *Alnus* vs *Quercus*, *Ambrosia* vs *Artemisia*,
*Ambrosia* vs *Fraxinus*, *Artemisia* vs *Fraxinus*, *Betula* vs *Phleum*, *Betula* vs *Quercus*, *Castanea* vs *Olea*, *Cedrus*
vs *Helianthus*, *Corylus* vs *Quercus*,  *Fagus* vs *Helianthus*, *Fagus* vs *Zea*, , and *Phleum* vs *Quercus*, all of which
show similar scattering intensities. Although it is not clear if the classification theory introduced above is
applicable to these groups, the theory should be applicable to other pairs as long as they have different scattering
intensity distributions.
The estimation of the pollen counts of *Alnus* and *Artemisia* had average errors of approximately 40% and 23%,
respectively. Test 4 had the largest error, with approximately 134% for *Alnus* and approximately 44% for
*Artemisia*, which increased the average error. It is difficult to identify an obvious reason for these large values,
but it is possible that the pollen samples were contaminated by dusts or pollen grains picked up for this experiment
were biased in size or shape.. Additionally, other estimations derived from the fitted curve of the front and the
side scattering distributions showed that even when the pollen counts are estimated only from scattering intensity
data without using total number of pollen grains, which is a known number, the pollen counts are able to be
calculated accurately. The KH-3000-01 has been widely used to estimate airborne concentrations of *Cryptomeria*
*japonica*. In this study, we found average errors of 20-40% for *Alnus* and *Artemisia*, values which are also likely
applicable to other taxa such as *Cryptomeria japonica*. Other taxa should, however, be investigated in future.
Pollen counts can be estimated by solving Eq. (5), which contains three equations, meaning that it is possible to
make estimates for three different pollen taxa simultaneously. If more integration intervals were picked up from
the probability density curve of the scattering intensity and added to the equation, in theory it would be possible
to count more pollen taxa. It is possible, however, that the accuracy of the estimated values might decline due to
the accuracy of the fitted curve. Therefore, narrowing down a target to two or three pollen types considering the
season should be helpful to make accurate automatic counts of several pollen taxa simultaneously.
In this study, the normal distribution function was chosen for fitting because of its universal property. However,
further consideration is required to determine the best function for fitting actual light scattering characteristics.

**5 Conclusion**

By applying the statistical analysis method, the Bonferroni method to the scattering patterns of *Alnus* at each time
step, our experiment showed that there seems to be no significant temporal changes in the light scattering patterns.
We also confirmed that different pollen types do not always have different light scattering patterns. However,
when two different pollen types have different light scattering patterns, it was possible to calculate the number of
pollen grains of each taxa using these light scattering patterns by solving the probability density function of the
pattern.

Code/Data availability: The authors confirm that the data supporting the findings of this study are available
within the article.

Author contributions: Kenji Miki established the system, performed the data analysis, and wrote the manuscript.
Shigeto Kawashima arranged the experimental setup and proofread the manuscript.
Conflict of interest: The authors declare that they have no conflict of interest.

Acknowledgement
This research was supported by the Young Research Exchange Programme between Japan and Switzerland
under the Japan-Swiss Science and Technology Programme.





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



Figure 1 Light scattering distribution data from *Alnus* and *Artemisia* used for estimation test.
Figure 2 Light scattering plots for *Alnus* pollen – fresh and after 1h, 2h, 6h, and 10 days storage.
Figure 3 Light scattering distribution of various pollen taxa.
Figure 4 Fitted curve for side scattering (top row) and probability density curve (second row) for *Alnus* (left)
and *Artemisia* (right) andfitted curve for front scattering (third row) and probability density curve (bottom row)
for *Alnus* (left) and *Artemisia* (right).
Figure 5 Results of automatic counting of *Alnus* and *Artemisia*. Red and black dots represent actual and
estimated numbers of pollen grains, respectively. The pair of red and black dots with the same shape are in the
same test set.

Table 1 Multiple comparison between *Alnus* data stored for various periods.
Table 2 Multiple comparison between each pollen taxon
Table 3 Results of estimation of number of pollen grains of *Alnus* and *Artemisia* and errors of each estimation.









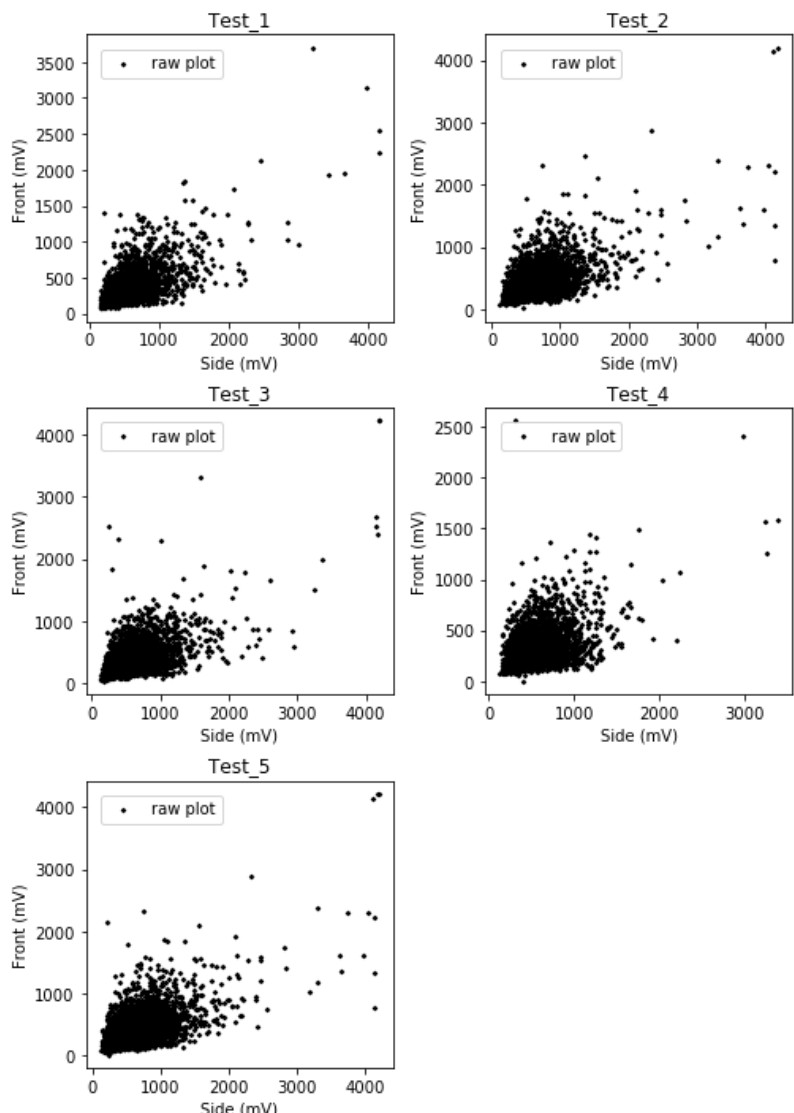


Fig.1                                                                                         Miki et al.







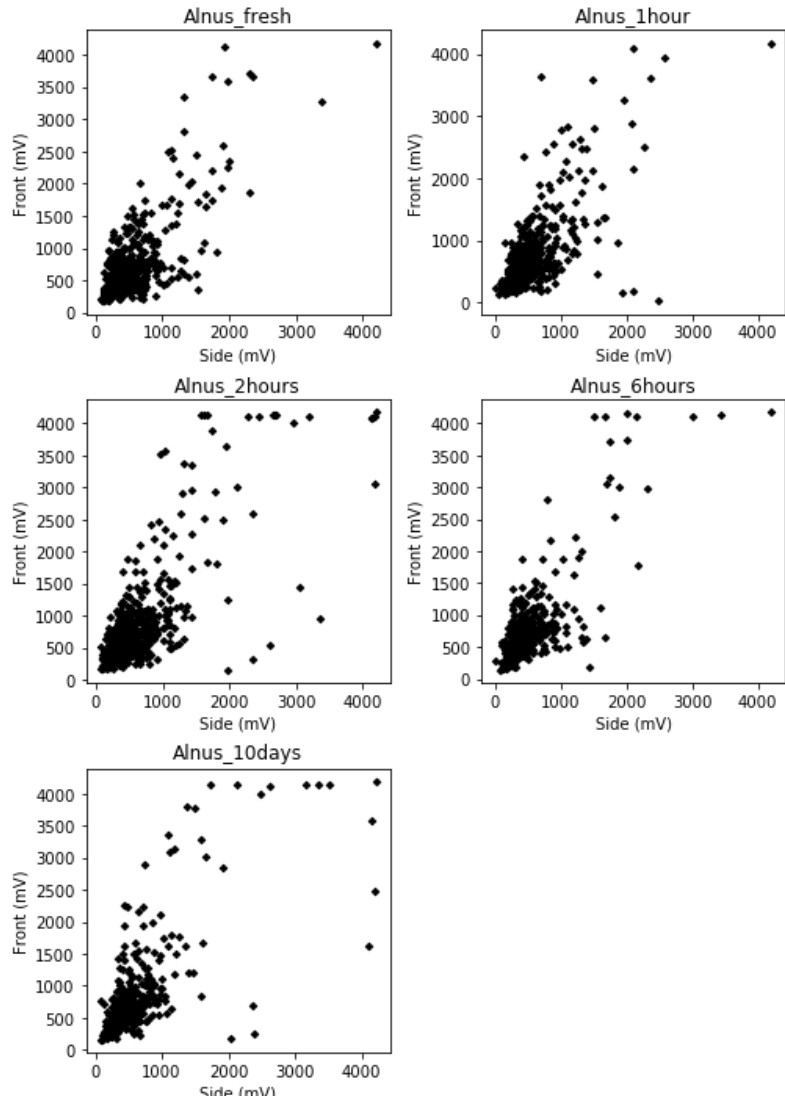







Fig.2                                              Miki et al.










Fig.3                                                                Miki et al.






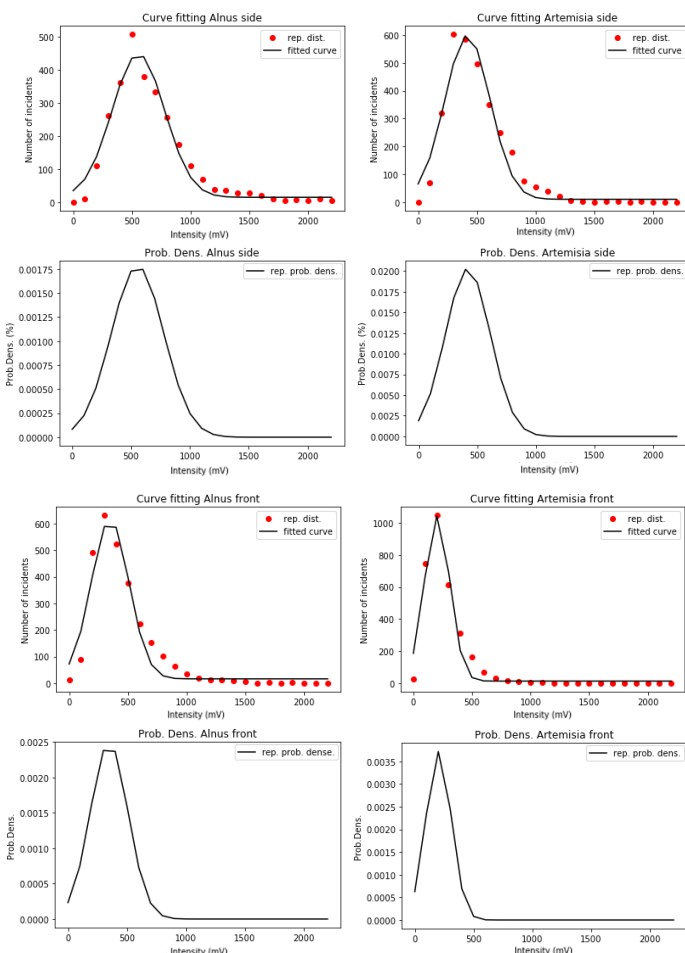


Fig.4                                                                                                    Miki et al.








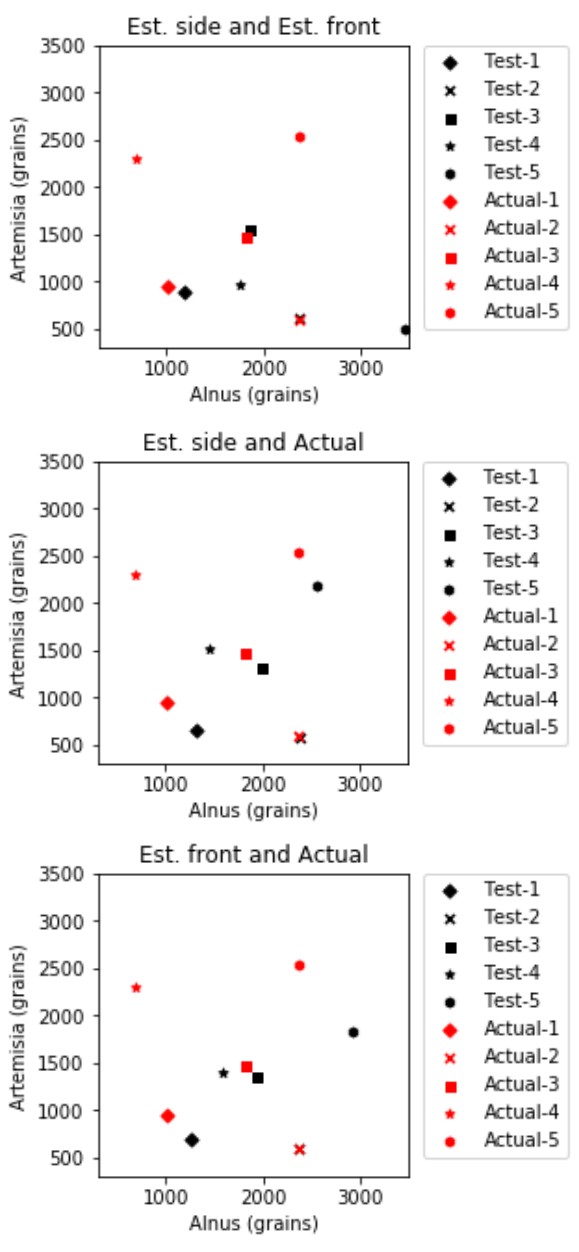



Fig.5                                              Miki et al.







Table 1  Multiple comparisons between each time step (*Alnus*)

| Side | 1hour | 2hour | 6hour | 10day |
|------|-------|-------|-------|-------|
| fresh | 1.00 | 0.38 | 1.00 | 1.00 |
| 1hour | — | 1.00 | 1.00 | 1.00 |
| 2hour | — | — | 0.71 | 1.00 |
| 6hour | — | — | — | 1.00 |

| Front | 1hour | 2hour | 6hour | 10day |
|-------|-------|-------|-------|-------|
| fresh | 1.00 | 1.00 | 1.00 | 1.00 |
| 1hour | — | 1.00 | 0.84 | 1.00 |
| 2hour | — | — | 1.00 | 1.00 |
| 6hour | — | — | — | 0.31 |




Miki et al.








Table 2 Multiple comparisons between each pollen taxon

Side

|  | Ambrosia | Artemisia | Betula | Castanea | Cedrus | Corylus | Fagus | Fraxinus | Helianthus | Olea | Phleum | Quercus | Zea |
|---|---|---|---|---|---|---|---|---|---|---|---|---|---|
| Alnus | 0.34 | * | * | * | * | 1.00 | * | * | * | 1.00 | * | * | * |
| Ambrosia | — | * | * | * | * | * | * | 0.08 | * | * | * | * | * |
| Artemisia | — | — | * | * | * | * | * | 0.06 | * | * | * | * | * |
| Betula | — | — | — | * | * | * | * | * | * | * | 0.06 | 1.00 | * |
| Castanea | — | — | — | — | * | * | * | * | * | * | * | * | * |
| Cedrus | — | — | — | — | — | * | * | * | * | * | * | * | * |
| Corylus | — | — | — | — | — | — | * | * | * | 0.49 | * | * | * |
| Fagus | — | — | — | — | — | — | — | * | * | * | * | * | 0.59 |
| Fraxinus | — | — | — | — | — | — | — | — | * | * | * | * | * |
| Helianthus | — | — | — | — | — | — | — | — | — | * | * | * | 1.00 |
| Olea | — | — | — | — | — | — | — | — | — | — | * | * | * |
| Phleum | — | — | — | — | — | — | — | — | — | — | — | 1.00 | * |
| Quercus | — | — | — | — | — | — | — | — | — | — | — | — | * |

* $p < 0.05$



Front

|  | Ambrosia | Artemisia | Betula | Castanea | Cedrus | Corylus | Fagus | Fraxinus | Helianthus | Olea | Phleum | Quercus | Zea |
|---|---|---|---|---|---|---|---|---|---|---|---|---|---|
| Alnus | * | * | * | * | * | 1.00 | * | * | * | * | * | 1.00 | * |
| Ambrosia | — | 0.95 | * | * | * | * | * | 1.00 | * | * | * | * | * |
| Artemisia | — | — | * | * | * | * | * | 1.00 | * | * | * | * | * |
| Betula | — | — | — | * | * | * | * | * | * | * | 1.00 | 1.00 | * |
| Castanea | — | — | — | — | * | * | * | * | * | 1.00 | * | * | * |
| Cedrus | — | — | — | — | — | * | * | * | 1.00 | * | * | * | * |
| Corylus | — | — | — | — | — | — | * | * | * | * | * | 1.00 | * |
| Fagus | — | — | — | — | — | — | — | * | 0.14 | * | * | * | 1.00 |
| Fraxinus | — | — | — | — | — | — | — | — | * | * | * | * | * |
| Helianthus | — | — | — | — | — | — | — | — | — | * | * | * | * |
| Olea | — | — | — | — | — | — | — | — | — | — | * | * | * |
| Phleum | — | — | — | — | — | — | — | — | — | — | — | 0.10 | * |
| Quercus | — | — | — | — | — | — | — | — | — | — | — | — | * |

* $p < 0.05$




Miki et al.









Table 3 Results of estimation of number of pollen grains of *Alnus* and *Artemisia* and errors of each estimation.

|  |  | Test 1 | | Test 2 | | Test 3 | |
|---|---|---|---|---|---|---|---|
|  |  | Alnus (error) | Artemisia (error) | Alnus (error) | Artemisia (error) | Alnus (error) | Artemisia (error) |
| Estimation | Side and Front | 1183 (17.36%) | 881 (6.77%) | 2367 (0.17%) | 612 (3.20%) | 1855 (1.76%) | 1552 (5.43%) |
|  | Total and Side | 1310 (29.96%) | 642 (32.06%) | 2386 (0.63%) | 577 (2.70%) | 1984 (8.83%) | 1310 (11.01%) |
|  | Total and Front | 1259 (24.90%) | 694 (26.56%) | 2378 (0.30%) | 585 (1.35%) | 1932(5.98%) | 1362 (7.47%) |
| Actual |  | 1008 | 945 | 2371 | 593 | 1823 | 1472 |


|  |  | Test 4 | | Test 5 | |
|---|---|---|---|---|---|
|  |  | Alnus | Artemisia | Alnus | Artemisia |
| Estimation | Side and Front | 1753 (157.42%) | 968 (57.86%) | 3469 (57.32%) | 489 (80.76%) |
|  | Total and Side | 1458 (114.10%) | 1520 (33.83%) | 2567 (16.42%) | 2179 (14.28%) |
|  | Total and Front | 1577 (131.57%) | 1402 (38.96%) | 2929 (32.83%) | 1817 (28.52%) |
| Actual |  | 681 | 2297 | 2205 | 2542 |









Miki et al.