# Peer review of "Estimation of pollen counts from light scattering intensity when sampling multiple pollen taxa —"

_Atmospheric Measurement Techniques, 2020_

## Referee Comment (RC1) · Anonymous Referee #2 · 24 Oct 2020

General Comments

Authors have developed a new method to spiciate the pollen type from the light scattering data. The method is potentially applied to the data obtained from the field measurements. The manuscript is clearly written and can be published after some minor revisions are made.

Specific comment Authors should provide the schematics for experimental setup. They

also should describe how the pollen grains were aerosolized in their study to sample them into the pollen counter.

It is more general to call "forward" scattering than "front" scattering.

It is helpful if authors can provide the approximate particles diameter range corresponding to the measured range of light scattering signals whose unit is volt.

Page 4, Line 136-137 P_Alnus_front should be changed to P_Artemisia_side

Probability density function shown in Figure 4 may be omitted as they have the same profile as the number of pollen incidents versus the voltage.

---

## Referee Comment (RC2) · Anonymous Referee #1 · 28 Oct 2020

Estimation of pollen counts from light scattering intensity when sampling multiple pollen taxa – Establishment of Automated Multi-taxa Pollen Counting Estimation System (AME System) is an excellent article

I agree totally with comments of Ref 2 and the answer of the authors I have nothing to add best regards

---

## Author Comment (AC2) · 30 Oct 2020

To Reviewer 1,

Thank you very much for your supportive comment.

Yours sincerely, Kenji Miki

---

## Author Response (AR1)

To Reviewer 1,

Estimation of pollen counts from light scattering intensity when sampling multiple pollen taxa – Establishment of Automated Multi-taxa Pollen Counting Estimation System (AME System) is an excellent article I agree totally with comments of Ref 2 and the answer of the authors I have nothing to add best regards

Thank you very much for your supportive comment.

To reviewer 2, Thank you very much for your very constructive advice and suggestions. I made

some correction following your comments.

Q1 Authors should provide the schematics for experimental setup.

A1 I added a figure C1 AMTD Interactive comment Printer-friendly version Discussion paper (Fig.1) to show the experimental setup.

Q2 It is more general to call "forward" scattering than "front" scattering.

A2 I changed C1 AMTD Interactive comment Printer-friendly version Discussion paper all "front" into "forward" in the draft.

Q3 It is helpful if authors can provide the approximate particles diameter range corresponding to the measured range of light scattering signals whose unit is volt.

A3 I totally agree that it will be very helpful if we would be able to provide the relationship between particle size and light scattering signals and it has been indeed a very important discussion point for the realisation of automated pollen counting system. However, the relationship between the strength of the signal in voltage is determined by multiple physical properties (size, roughness, etc.) of sampled particle as discussed in Matsuda and Kawashima (2018), so it might be misleading to generalise the relationship between the particle diameter and signal strength. I added some sentences explaining this to add more clearness.

Q4 Page 4, Line 136-137 P_Alnus_front should be changed to P_Artemisia_side

A4 I made the revision following the advice.

Q5 Probability density function shown in Figure 4 may be omitted as they have the same profile as the number of pollen incidents versus the voltage.

A5 I deleted the figures of the probability densities following your advice. Thank you very much for your advice again.

[Figure]

Fig. 1                                                                 Miki and Kawashim

---

## Author Response (AR2)

Dr. Pierre Herckes,

Thank you for your comments and advice.
I made some correction following your advice. I hope the correction satisfies the journal criteria.

Q.1
Could you please add error bars to figures.
A.1
Each value in figure 6 represents one test result respectively. So, the error bar is not applicable in this case because error bars can be applicable when each value represents plural results.

Q.2 Figure 2: Could you please delete the "raw data" legend as it is repetitive and clear from the legend. It seems to obscure some data points too.

A. 2 I deleted the label following your advice.

Q.3 Figure 6 seems to have low resolution as well as some others. Please work with the editorial office to ensure high quality figures.

A.3 I managed to make figures higher resolution.
I replaced Fig.2-6 with the higher resolution figures.

Q. 4 Please use only significant digits in your manuscript. E.g. table 3, please write 57% error not 57.32%
A.4 I made corrections following your advice.

Thank you again for your advice.
Sincerely yours,
Kenji MIKI